# High temperature sensitivity of Arctic isoprene emissions explained by sedges

Hui Wang [1] ✉, Allison M. Welch[1], Sanjeevi Nagalingam [1], Christopher Leong[1], Claudia I. Czimczik [1], Jing Tang [2], Roger Seco [3], Riikka Rinnan [2] ✉, Lejish Vettikkat [4], Siegfried Schobesberger [4], Thomas Holst[5], Shobhit Brijesh[1], Rebecca J. Sheesley [6], Kelley C. Barsanti[7,8] & Alex B. Guenther [1] ✉

It has been widely reported that isoprene emissions from the Arctic ecosystem have a strong temperature response. Here we identify sedges (*Carex* spp. and *Eriophorum* spp.) as key contributors to this high sensitivity using plant chamber experiments. We observe that sedges exhibit a markedly stronger temperature response compared to that of other isoprene emitters and predictions by the widely accepted isoprene emission model, the Model of Emissions of Gases and Aerosols from Nature (MEGAN). MEGAN is able to reproduce eddy-covariance flux observations at three high-latitude sites by integrating our findings. Furthermore, the omission of the strong temperature responses of Arctic isoprene emitters causes a 20% underestimation of isoprene emissions for the high-latitude regions of the Northern Hemisphere during 2000-2009 in the Community Land Model with the MEGAN scheme. We also find that the existing model had underestimated the long-term trend of isoprene emissions from 1960 to 2009 by 55% for the high-latitude regions.

Rapid climate change in the Arctic is strongly influencing terrestrial ecosystems[1,2]. The change in both climate and ecosystems could alter the atmospheric chemistry and composition in the Arctic atmosphere through biogenic volatile organic compounds (BVOCs) emitted by plants[3–5]. Since the Arctic has limited anthropogenic VOC sources, BVOCs have a key role in high-latitude atmospheric chemistry[6]. Because BVOCs are the main precursors of secondary organic aerosol (SOA)[7], changes in BVOC emissions will likely affect the quantity and characteristics of SOA and, thus the climate system in the Arctic[8–11]. Furthermore, the rise in BVOC levels could decrease the atmospheric oxidation capacity and prolong the lifetime of methane, thereby exacerbating global warming[11,12].

Isoprene is the most abundant reactive BVOC emitted globally and in the Arctic[3,13,14]. Isoprene can help vegetation tolerate abiotic stresses[15], and isoprene can act as a signaling compound to stimulate plant defense mechanisms during stress periods[16]. Isoprene is synthesized from dimethylallyl diphosphate (DMADP) derived from the methyl erythritol 4-phosphate (MEP) pathway through the enzyme isoprene synthase (IspS)[17]. Isoprene emission is controlled by environmental conditions, especially temperature and solar radiation[18]. Thus, a rapidly warming climate in the Arctic is favorable for increasing the emission of isoprene[19–22]. The temperature response curves of isoprene emission, used in the current earth system models (ESMs) and the chemistry transport models (CTMs), are based on measurements of a few temperate plants[13,23], and a typical isoprene

[1]Department of Earth System Science, University of California, Irvine, California, USA. [2]Center of Volatile Interactions (VOLT), Department of Biology, University of Copenhagen, København, Denmark. [3]Institute of Environmental Assessment and Water Research (IDAEA-CSIC), Barcelona, Catalonia, Spain. [4]Department of Technical Physics, University of Eastern Finland, Kuopio, Finland. [5]Department of Physical Geography and Ecosystem Science, Lund University, Lund, Sweden. [6]Department of Environmental Science, Baylor University, Waco, Texas, USA. [7]Department of Chemical & Environmental Engineering, Center for Environmental Research & Technology, University of California Riverside, Riverside, California, USA. [8]Atmospheric Chemistry Observations and Modeling Laboratory, National Center for Atmospheric Research, Boulder, Colorado, USA. ✉e-mail: huiw16@uci.edu; riikkar@bio.ku.dk; alex.guenther@uci.edu

temperature response curve has a Q10 of about 3, which is thought to be driven by the influence of temperature on substrate supply and the activity of IspS[17]. However, recent whole-ecosystem measurements suggest that the temperature response of isoprene emissions in high-latitude tundra ecosystems has a Q10 over 8, which is also much higher than that predicted by the widely used BVOC emission model, the Model of Emissions of Gases and Aerosols from Nature (MEGAN)[21,24–28,29]. In contrast, leaf/branch-level studies showed that the Arctic willow species (*Salix pulchra*, *Salix glauca*, and *Salix myrsinites*), which are one of main isoprene emitters in high-latitude tundra ecosystems, have a short-term temperature response that is similar to temperate plants[25,30]. Our previous study also confirmed that the hourly temperature response curve of *Salix* spp. is consistent with that of temperate plants as well as the MEGAN model[13,31]. Additionally, we found that the isoprene emission factors of willows show a greater-than-expected response to the mean ambient temperature of the previous day[31]. Nonetheless, we concluded that the temperature response of willows in the Arctic is greater than that predicted by current models but still cannot fully explain the high-temperature sensitivity of isoprene emissions from high-latitude ecosystems[31]. Consequently, species-specific investigations are necessary to further explore the strong temperature responses observed at the ecosystem level.

In this study, we identified sedges (*Carex* spp. and *Eriophorum* spp.) as the key contributors to the pronounced temperature sensitivity of isoprene emissions in the Arctic ecosystems. Sedges exhibit a more significant temperature response than both willows and MEGAN. Additionally, we observed that sedges can adjust their temperature sensitivity and emission capacity, which is represented by the emission factor in this paper, in response to changes in ambient growth temperatures. We integrated these findings into the MEGANv2.1 model[13],

enhancing its capability to simulate observed ecosystem-scale flux measurements. Moreover, the updated MEGAN model projects a 20% increase in Arctic isoprene emissions and a 55% rise in the long-term isoprene emission trend. Given the ongoing intensification of global warming, the changes in isoprene emissions have the potential to substantially alter atmospheric chemistry in high-latitude regions.

## Results and discussion
### Isoprene from sedges is sensitive to temperature changes
Our species-level gas exchange chamber experiments showed that the main isoprene emitters among the species measured at Toolik Field Station (TFS), Alaska, USA, are sedges, a major component of Arctic graminoid plants, and willows, a major component of Arctic woody plants (Supplementary Fig. 1). Other studies have also indicated that the *Salix* spp., *Carex* spp. and *Eriophorum* spp. exhibit significantly higher isoprene emission levels compared to other tundra species, e.g., *Betula* spp. and *Cassiope* spp[25,30,32–34]. and *Sphagnum* spp[35–37]. The temperature response of willows in the Arctic cannot explain the high-temperature sensitivity of isoprene emissions from high-latitude ecosystems[31].

Arctic sedges studied here show a more pronounced temperature response than other plant species, including Arctic woody willow shrubs and any of the plant responses used to develop the MEGAN model. Our data confirm that the sedges are responsible for the heightened temperature responses of isoprene emissions from high-latitude ecosystems (Fig. 1a and 1b). We calculated the Q10 coefficient for isoprene emissions from sedges (*Carex* spp. and *Eriophorum* spp.) and willows between 25 and 35 °C. The Q10 coefficient represents the isoprene emission rate change with a 10 °C rise in the leaf temperature. The Q10 values of *Carex* spp. (15.6 ± 8.8) and *Eriophorum* spp. (9.1 ± 7.0) are much higher than the Q10 of the Arctic willows (3.2 ± 1.8), which is

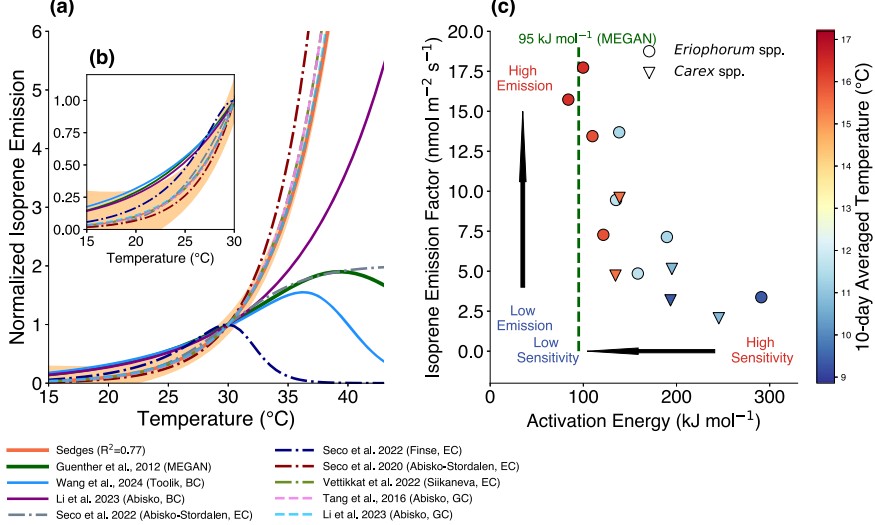

**Fig. 1 | Isoprene temperature sensitivities from sedges and other measurements. a** presents the temperature responses of isoprene emissions from this and previous studies in the northern high-latitude regions. (**b**) is the same plot as (**a**), but only for temperatures under 30 °C; (**c**) shows the relationship between the isoprene temperature sensitivities and emission capacities of sedges. (**a**) The short-term temperature response curves of sedges up to 35 °C from this study are shown by the orange solid line, and the orange shading represents the 95% confidence intervals. The short-term temperature response curves of tundra ecosystem from previous studies are also presented by lines with different colors and patterns. GC, BC and EC represent ground chamber experiments, branch chamber experiments and eddy-covariance measurements. The temperature response curves are normalized to the emission level when the leaf temperature equals 30 °C. The temperature curves in Tang, et al. [29]. and Li, et al. [25]. came from the ground chamber observations of mixed local vegetation at the Abisko site. Li, et al. [25]. also did the

branch chamber experiments for *Salix myrsinites* L. (purple solid line). The site in Seco, et al. [21] is located in a sedge-dominated fen near the Abisko-Stordalen site. The Abisko measurements in Seco, et al. [24]. happened at a different location within the same Abisko-Stordalen area on an ombrotrophic permafrost plateau. The Finse site in Seco, et al. [24]. is a tundra with a mixture of fen and heath vegetation with shrubs and lichens. The Siikaneva site is in a fen dominated by moss, sedges and dwarf shrubs, and surrounded by Scots pine forest[26]. **c** presents an inverse relationship between the activation energies of the isoprene temperature response and the isoprene emission factors for *Eriophorum* spp. (circle) and *Carex* spp. (triangle). The green dashed line in (**c**) shows the activation energy in the Model of Emissions of Gases and Aerosols from Nature (MEGAN)[13]. The colors of markers denote the average temperatures over the previous 10 days, and the emission factor is defined as the level of isoprene emission at a leaf temperature of 30 °C and a photosynthetic photon flux density (PPFD) of 1000 μmol m⁻² s⁻¹.

close to the Q10 of the MEGAN model (2.91) (Supplementary Fig. 2). We applied the Arrhenius equation to model the exponential temperature response curves of *Carex* spp. and *Eriophorum* spp. (refer to the Methods "section"), where the activation energy (Eq. 5) denotes the temperature sensitivity of isoprene emission. Our findings indicate that the temperature response curves of both *Carex* spp. and *Eriophorum* spp. exhibit high-temperature sensitivity (or high activation energy) up to 35 °C (Supplementary Fig. 3). However, the activation energy and $R^2$ decrease beyond 40 °C, suggesting a slower increase rate of isoprene emissions from both species (Supplementary Fig. 3).

Moreover, there is an inverse relationship between the temperature sensitivity and emission capacity (or emission factor) of isoprene in sedges, both of which are acclimated to the temperature history of the previous days (Fig. 1c). We used the 10-day average temperature as an indicator of the recent growing environment. This choice is based on the Pearson correlation coefficient for the temperature sensitivities (activation energy in Eq. 5) and emission factors in relation to the mean temperature of the preceding 1 to 15 days for *Eriophorum* spp. (Supplementary Fig. 4). The use of a 10-day average temperature also follows the current framework of the MEGAN model but the actual time period influencing isoprene emission is uncertain. With higher 10-day average temperatures, *Carex* spp. and *Eriophorum* spp. exhibit lower temperature sensitivity of isoprene emissions, as indicated by a decrease in activation energy, and higher emission capacity, or emission factors, compared to those in colder environments. Isoprene emissions are controlled by the enzyme activity and supply of substrates[17]. In addition to these two factors, we speculate that the pronounced response of sedges to high temperatures could also be related to enzyme accumulation. The mRNA levels and IspS protein concentrations are likely low in sedges in a cold environment until a warm environment or a specific temperature threshold initiates the gene expression necessary for IspS synthesis. The IspS protein has a half-life time of 5.3 days in a 20 °C environment[38]. Its accumulation, following the onset of gene expression for IspS synthesis, would result in increased isoprene emissions as conditions transition from cold to warm. When the basal level IspS reaches the maximum, isoprene emission would become primarily dependent on enzyme activity and substrate availability[17,39]. One potential piece of evidence is that the isoprene activation energy of *Eriophorum* spp. after a warming period is comparable to that of aspen (*Populus* spp.) at 72.1 kJ mol⁻¹ [39] and that of MEGAN at 95 kJ mol⁻¹ [13], suggesting that IspS levels are no longer limiting factors, and the emission patterns resemble those of temperate isoprene emitters (Fig. 1c). It has been reported that the isoprene emission of the Arctic sedges varies with the air temperature[40,41], and our results provide evidence that the sedges are sensitive to both minutes-hour scale temperature change as well as day-weeks change.

## Updated MEGAN model can better explain the flux measurements

The MEGANv2.1 model, which has 16 plant functional types (PFTs) each with a uniform temperature response curve, cannot fully capture the observed variability in isoprene emissions in Arctic regions. To address this, we updated the temperature response curves in MEGAN version 2.1[13,42] and evaluated the model using eddy-covariance flux measurements from three high-latitude sites. Supplementary Table 1 shows that sedges are the dominant isoprene emitters at the Abisko-Stordalen and Siikaneva sites, while the Finse site is dominated by both sedges and willows. In this study, the PFT categories of Arctic grass and boreal broadleaf deciduous shrubs are updated based on our measurements.

The Arctic grass adopted the temperature curve of sedges (Eq. 5 in the Methods) with a dynamic activation energy (Eq. 6 in the Methods) and emission factor (Eq. 7 in the Methods) derived from *Eriophorum* spp. measurements in this study (Supplementary Fig. 5). The boreal broadleaf deciduous shrub used the temperature response curve of

MEGANv2.1 (Eqs. 2, 3, and 4 in the Methods) but included the response of emission factors to the previous 1-day average temperature for *Salix* spp. (Eq. 8 in the Methods)[31]. The leaf-level emission factor was 12.0 nmol m⁻² s⁻¹ for the Arctic grass and 6.5 nmol m⁻² s⁻¹ for the boreal shrub (including willows), respectively, which was based on our glass chamber measurements in this study and Wang, et al.[31]. Besides temperature response curves, isoprene emissions at the field sites were also affected by the abundance of sedge and willow. To eliminate uncertainties associated with vegetation fraction input, we used the least squares fitting method to adjust vegetation proportions, thereby optimizing the performance of the model in comparison with flux measurements.

The updated model demonstrates an enhanced ability to capture the variability of isoprene flux compared to the default temperature response curve of the MEGANv2.1 model (see Fig. 2 and Supplementary Table 2). Across the three sites, the updated model consistently shows an increase in $R^2$ and a decrease in Root Mean Square Error (RMSE). The updated model can capture both the low values during cold periods and the high values in warm periods more accurately than the default model. In addition, for the Siikaneva site, the model accurately captures the elevated values following heatwaves, after accounting for the effects of warming on temperature response and emission factors. We also compared the fitted fractions for Arctic grass and boreal shrub from the updated model with the observed land cover/vegetation fractions at the Finse and Siikaneva sites (see Table 1). Our findings indicate that the fitted fractions for Arctic grass and boreal shrub closely match the observations or estimates at Finse and Siikaneva sites.

## Estimation of high-latitude isoprene emission

We reported a pronounced temperature response curve for sedges in this study. In addition, our previous research indicated that Arctic willows (*Salix* spp.) can significantly increase their isoprene emission capacity in response to an increase in the average temperature of the preceding day[31]. The different isoprene emission patterns of the two types of high-latitude plants (Arctic grass and boreal shrub) both point to an increase in isoprene emissions due to Arctic warming. To estimate isoprene emissions in the high latitude regions (north of 60°N), we updated MEGANv2.1 in the Community Land Model version 5 (CLM5)[43] with the temperature curves from this study for the Arctic grass and boreal broadleaf deciduous shrub PFTs. We used the emission factors reported by Guenther, et al.[13]. for the CLM simulations. The averaged isoprene emissions estimated by the updated model are 20% higher than the original MEGAN estimate for 2000-2009, increasing from 2.71 to 3.25 Tg yr⁻¹, which suggests that the isoprene emissions from the high-latitude regions in the Northern Hemisphere are underestimated in current ESMs. By updating the temperature response, the CLM simulation shows a decrease in isoprene emissions from Arctic grass-dominated tundra, while observing an increase from high-latitude deciduous shrubs (Fig. 3 and Supplementary Fig. 6). In the updated model, Arctic grass isoprene emissions respond less to low temperatures typical of Arctic climates but increase significantly with rising temperatures compared to the default model (Fig.1 and Fig. 3). The simulations predict a notable increase in isoprene emissions in the Russian Siberian regions dominated by boreal deciduous shrubs. The model results presented by Stavrakou, et al.[44]. suggested that the interannual variability of Ozone Monitoring Instrument (OMI) formaldehyde (HCHO) measurements in Siberia could be explained by biogenic isoprene emissions. However, more in-situ measurements are crucial to validate the model and understand the biogenic isoprene emissions in Siberia.

Additionally, we calculated the long-term trend of isoprene emissions and found that the updated temperature response resulted in a 55% increase in the trend of isoprene emissions from 1960 to 2009 in the high-latitude region compared to the default model (Fig. 4). The

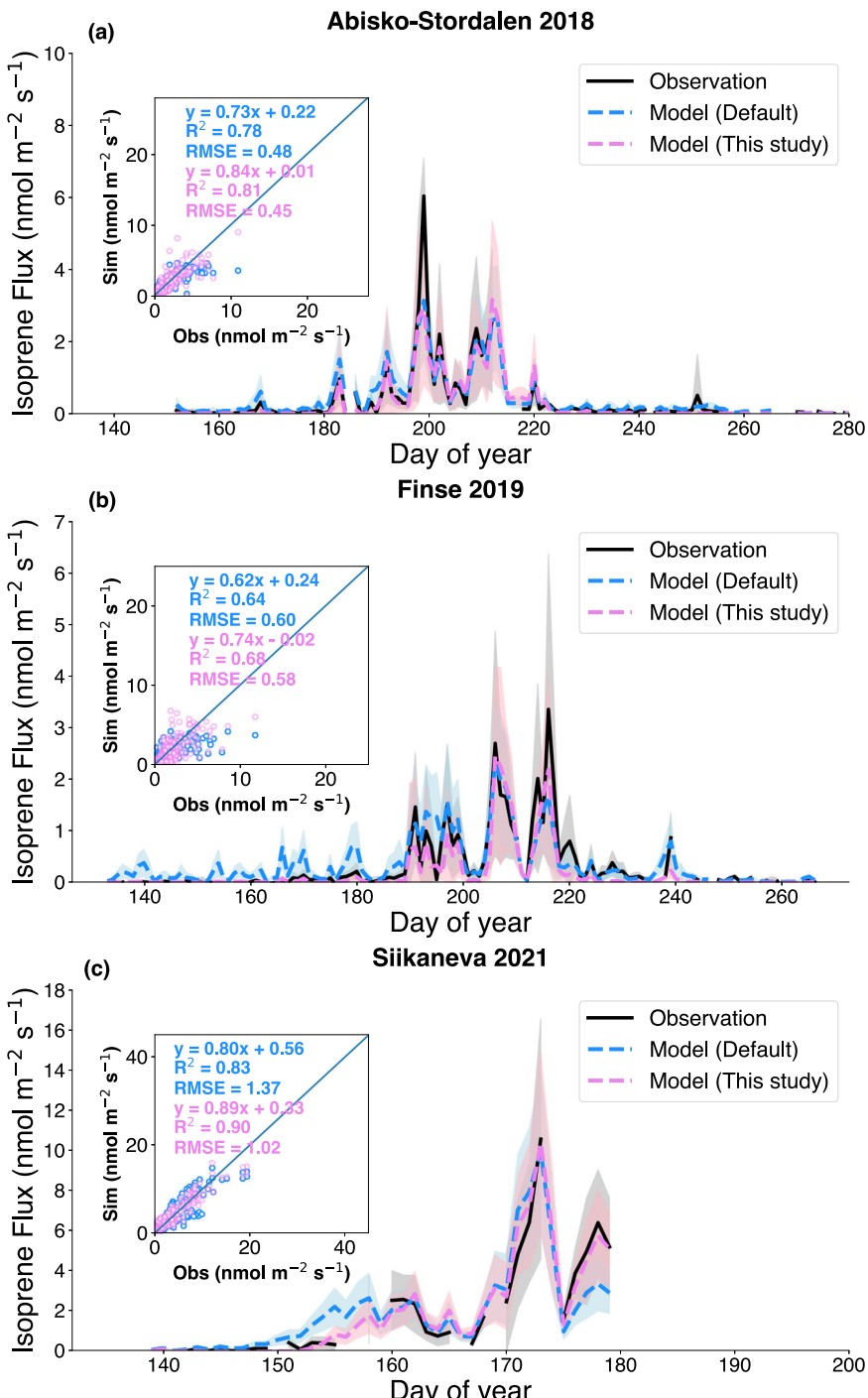

**Fig. 2 | Time series of the daily observed and simulated isoprene flux by MEGAN with the default (blue) and updated (pink) temperature responses.** The eddy-covariance flux measurements (black line) from three high-latitude sites, Abisko-Stordalen site in 2018[24] (**a**), Finse site in 2019[24] (**b**) and the Siikaneva site in 2021[26] (**c**), were evaluated. The shaded areas in different colors represent the standard deviation of the daily fluxes from observations (gray), the default MEGAN (blue), and the updated MEGAN (pink), respectively. The scatter plots illustrate the performance of the models compared to half-hourly isoprene flux measurements. Isoprene measurements with a photosynthetic photon flux density exceeding 300 μmol m$^{-2}$ s$^{-1}$ were taken for comparison. The updated model incorporates Arctic grass and boreal shrub temperature response curves, while the default model uses the temperature curves in MEGAN v2.1[13].

default model framework could not capture the rapid change in high-latitude isoprene emissions and their feedback on atmospheric chemistry and the climate system. Furthermore, the results showed that an Arctic heatwave could create a significant isoprene burst event. For example, abnormally warm Arctic weather in 1991 and 2001 was predicted to have an ~40% increase in isoprene emissions in high-latitude regions in the Northern Hemisphere (Supplementary Fig. 7).

Increased heatwave frequency[45] and general warming could intensify high-latitude isoprene impacts, with significant shifts in isoprene emissions potentially altering local atmospheric chemistry and climate dynamics. The isoprene emitters, including sedges and willows[31], would respond to both short-term, intense heatwaves and long-term warming from minutes to weeks scales by increasing their isoprene emissions (Fig. 5). The lower atmospheric oxidative

**Table. 1 | The cover fractions of plant functional types that were estimated by the model and land survey at the Finse, Abisko-Stordalen, and Siikaneva sites**

|  | Fitted sedge fraction (%) | Observed/Estimated sedge fraction (%) | Fitted shrub fraction (%) | Observed/Estimated shrub fraction (%) | Reference |
|---|---|---|---|---|---|
| Abisko-Stordalen | 11.1 | - | - | - | - |
| Finse | 13.3 | 17.2 | 7.4 | 6.3 | Ramtvedt[52] |
| Siikaneva | 26.8 | 24.0 | - | - | Vettikkat, et al.[26] |

The cover fractions of sedges and isoprene-emitting shrubs in the model are fitted using the least square methods.

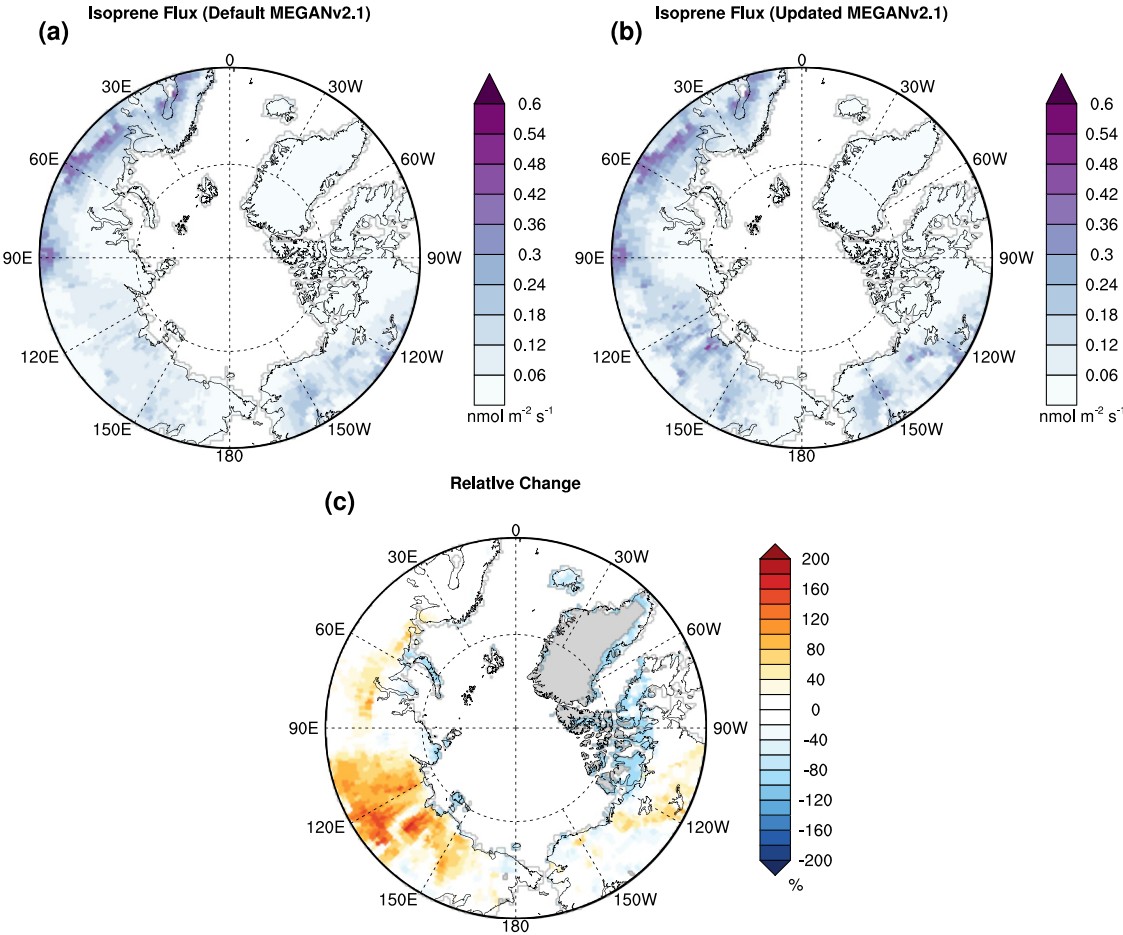

**Fig. 3 | The averaged isoprene emissions in high-latitude regions (north of 60°N) during 2000–2009 estimated by MEGAN.** The default MEGAN (**a**) and the updated MEGAN (**b**) were driven by the CLM5 model. The relative change caused by the updated temperature response curves is presented in (**c**). The averaged isoprene emissions estimated by the updated model are 3.25 Tg yr$^{-1}$, around 20 % higher than the original MEGAN estimate of 2.71 Tg yr$^{-1}$.

capacity caused by rising isoprene emissions could further increase the lifetime of methane and then exacerbate warming. Boy, et al.[12] suggest that a 6 °C warming could increase the lifetime of methane by 11.4% at a boreal site dominated by monoterpene emissions. The pronounced response of isoprene to warming suggested by this study will exacerbate the BVOC–OH–CH$_4$ feedback. Additionally, the increased isoprene emissions could also disturb aerosol[46] and ozone formation, as well as aerosol-cloud interactions[10,11]. BVOCs, including isoprene, act as sinks for tropospheric ozone in the Arctic[47], and the increase of isoprene could further diminish tropospheric ozone. However, the increased frequency and intensity of wildfire and anthropogenic emissions from more ship activities[48–50], the atmospheric transport of NO$_x$ (NO + NO$_2$) and peroxyacetyl nitrate could alter the chemical regime and change the tropospheric ozone and aerosol formation.

## Methods

### BEARS-oNS campaign and Glass Chamber Experiments

The Biogenic Emissions and Aerosol Response on the North Slope (BEAR-oNS) project aimed to explore the impact of climate change on the interactions among BVOC, aerosol, and climate in high-latitude regions. The BEAR-oNS campaign was conducted during July and early August of 2022 and 2023 to characterize the BVOC emission and investigate the source of aerosols in the high-latitude tundra ecosystem near the Toolik Field Station (TFS) in Alaska (USA, 68.65 N,149.58 W). July mean air temperatures at TFS were 9.6 °C in 2022 and 13.1 °C in 2023, with accumulated precipitation at 220 mm and 105 mm for each year respectively.

The temperature response curve experiments investigated the impact of temperature on isoprene emission. The vegetation specimens analyzed in this study are listed in Supplementary Table 3 and

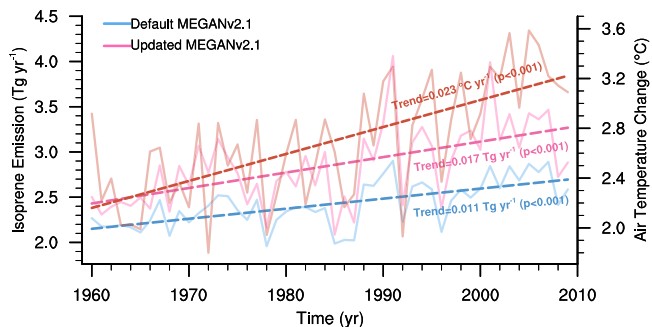

**Fig. 4 | Long-term trend of isoprene emission in high-latitude regions (north of 60°N) estimated by MEGAN during 1960–2009.** Time series of isoprene emissions, as estimated by the default MEGAN and the updated MEGAN, are depicted by the blue and pink solid lines, respectively. Changes in air temperature over land are shown by the orange solid line. The linear trends of isoprene emissions, as estimated by the default MEGAN and the updated MEGANv2.1, are indicated by the blue and pink dashed lines, respectively. The linear trend of air temperature over land is represented by the orange dashed line. The significance of the linear trends for isoprene emissions and air temperature was tested using the Mann-Kendall test. For high latitude regions (north of 60°N), the linear trends in isoprene emissions are estimated to be 0.017 Tg yr$^{-1}$ ($p < 0.001$) for the updated MEGANv2.1 and 0.011 Tg yr$^{-1}$ ($p < 0.001$) for the default MEGANv2.1.

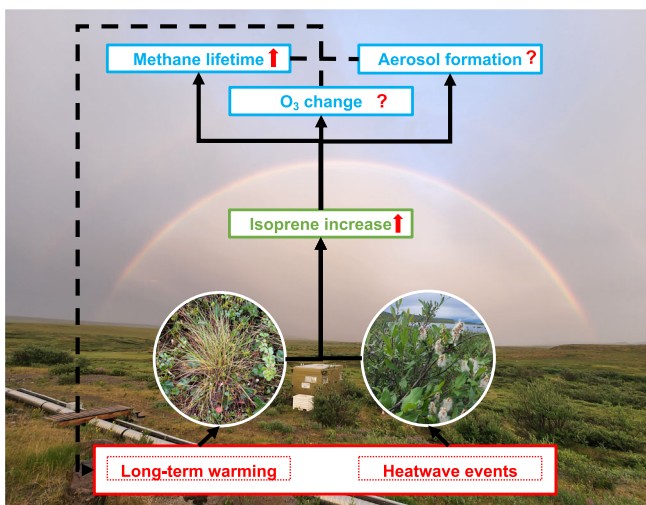

**Fig. 5 | Schematic representation of isoprene emission increase and atmospheric chemical changes induced by the response of sedges and willows to Arctic warming.** Warming will increase isoprene emissions from Arctic ecosystems. This change in isoprene emissions due to warming can alter tropospheric chemistry, resulting in extended methane lifetime, altered tropospheric ozone concentration, and changed aerosol formation. These changes can influence the local radiation energy balance and exacerbate climate fluctuations.

Supplementary Table 4. The vegetation samples were collected from the tussock tundra near the Toolik Field Station. The plant samples were detached from soil with the main root system or cut from the main branch and submerged in water in glass bottles. We chose plants that were in good condition without galls or visible damage.

The temperature curve experiments were conducted with a leaf chamber system (see details in the next section). One chamber blank (background) VOC sample from an empty chamber was collected at the beginning of every experiment before putting the plant into the glass chamber. After placing a leaf, branch, or glass dish with moss into the glass chamber, VOC sampling was initiated after the photosynthesis rate stabilized. For *Carex1* and *Eriophorum1* samples, there were

only two temperature steps 20 °C and 30 °C, collected after the photosynthesis rate was relatively stable. For other sedge samples, the leaf temperature was ramped up from 15 or 20 °C to 35 or 40 °C in steps of 5 °C. Each temperature step lasted 1 hour, and samples were collected with sorbent cartridges (see next section) during the last 10 to 15 minutes of the hour with a flow rate of 200 cc min$^{-1}$ for 5 min resulting in a 1 L VOC sample.

## Leaf chamber

The leaf-level BVOC measurements were conducted using a custom-made, field-portable glass chamber with environmental controls. The chamber has an internal volume of 0.62 L and is mounted on a thermoelectric cooler assembly (Custom Thermoelectric, MD, USA), which allows for precise control of the leaf temperature. A miniature fan was installed to stir the air inside the chamber. A white LED source provided artificial illumination with a photosynthetically active radiation output of ~1000 μmol m$^{-2}$ s$^{-1}$. The ambient air was pushed into the chamber using a diaphragm pump at rates of 0.9–1.0 L min$^{-1}$, and the VOCs in the inlet air flow were removed by an activated carbon filter. Part of the effluent air from the chamber was sampled onto sorbent cartridges (Tenax TA and Carbograph 5TD; Markes International, UK) for BVOC analysis. Additionally, an infrared gas analyzer (LI-850; LI-COR Biosciences, NE, USA) was used to measure the $CO_2$ and $H_2O$ mixing ratios in the influent (background) and effluent airflows; the analyzer was switched between the chamber's inlet and outlet every 30 s.

## Gas chromatography system

The sampled sorbent cartridges were transported to our laboratory at the University of California, Irvine, where they were thermally desorbed using a TD autosampler (Ultra-xr; Markes International). The desorbed VOCs were injected into a gas chromatograph (GC) (7890B; Agilent Technologies, CA, USA) equipped with a 60 m Rxi-624Sil MS capillary column (Restek, PA, USA). The column eluate was channeled to an electron impact ionization time-of-flight mass spectrometer (BenchTOF-Select; Markes International) and a flame ionization detector (FID, Agilent) for compound identification and quantification. A detailed explanation of the GC methodology including the GC oven temperature program, calibration protocols, and measurement uncertainties, is provided elsewhere[51].

## MEGAN model and temperature response curve

MEGAN is a flexible model framework for calculating biogenic volatile organic compound emissions from terrestrial ecosystems[13,23]. The isoprene flux in MEGANv2.1 is calculated as:

$$F = \varepsilon \cdot LAI \cdot Cce \cdot \gamma_T \cdot \gamma_P \cdot \gamma_A \cdot \gamma_C \cdot \gamma_{SM} \quad (1)$$

where $\varepsilon$, $Cce$, and LAI represent the canopy-level standard emission factor (nmol m$^{-2}$ s$^{-1}$), canopy factor ($= 0.3$), leaf area index (LAI, m$^2$ m$^{-2}$). $\gamma_T$, $\gamma_P$, $\gamma_A$, $\gamma_C$ and $\gamma_{SM}$ denote the emission activity factors of isoprene emissions including temperature, solar radiation, leaf age, $CO_2$ inhibition and water stress, respectively. MEGAN considers the BVOC responses to the long-term temperature, defined as the temperature of the past one day or longer, and the current temperature, reflecting changes on a minute-to-hour scale. The default short-term temperature response curve, $\gamma_T$, for isoprene in MEGAN is:

$$\gamma_T = E_{opt} \cdot \frac{C_{T2} \cdot e^{\frac{C_{T1}}{R}\left(\frac{1}{T_{opt}} - \frac{1}{T}\right)}}{C_{T2} - C_{T1} \cdot \left(1 - e^{\frac{C_{T2}}{R}\left(\frac{1}{T_{opt}} - \frac{1}{T}\right)}\right)} \quad (2)$$

$T$ (K) is the leaf temperature. R ($= 0.008314$ kJ mol$^{-1}$), $C_{T1}$ ($= 95$ kJ mol$^{-1}$) and $C_{T2}$ ($= 230$ kJ mol$^{-1}$) are the gas constant and the

activation and deactivation energies, respectively. $E_{opt}$ and $T_{opt}$ are:

$$T_{opt} = 313 + 0.6 \cdot (T_{240} - 297.15) \tag{3}$$

$$E_{opt} = 2 \cdot e^{0.05 \cdot (T_{24} - 297.15)} \cdot e^{0.05 \cdot (T_{240} - 297.15)} \tag{4}$$

where $T_{24}$ (K) and $T_{240}$ (K) denote the mean air temperature of the previous 24 hours and 240 hours, respectively. Equations (3) and (4) for $T_{opt}$ (K) and $E_{opt}$ represent the long-term impact of temperature on the optimal temperature and the shape of the temperature response curve.

The updated short-term temperature response curve for sedges in MEGANv2.1 is expressed as:

$$\gamma_{T_{sg}} = E_{opt\_sg} \cdot e^{\frac{C_{sg}}{R}\left(\frac{1}{303.15} - \frac{1}{T}\right)} \tag{5}$$

$C_{sg}$ is the activation energy for the isoprene temperature response of sedges and changes with $T_{240}$ as:

$$C_{sg} = 95 + 9.5 \cdot e^{0.53 \cdot (288.15 - T_{240})} \tag{6}$$

The impact of $T_{240}$ on isoprene emission factor of sedge is as:

$$E_{opt\_sg} = e^{0.12 \cdot (T_{240} - 288.15)} \tag{7}$$

The PFT of boreal broadleaf deciduous shrub in the updated model adopted the temperature response curve described in the Eqs. (2)–(4) with the parameters in Wang et al. (2023), and the long-term temperature response for boreal broadleaf deciduous shrub in MEGANv2.1 is updated as:

$$E_{opt\_willow} = 7.9 \cdot e^{0.22 \cdot (T_{24} - 297.15)} \tag{8}$$

To investigate the capacity of different temperature response curves to explain eddy covariance measurements, we rewrote Eq. (1) as:

$$F_{flux} = \sum_{i=1}^{n} E_i \cdot \frac{LAI}{LAI_{max}} \cdot \gamma_{T\_i} \cdot \gamma_{others} \tag{9}$$

where $\gamma_{others}$ represents the product of $C_{ce}$, $\gamma_P$, $\gamma_A$, $\gamma_C$ and $\gamma_{SM}$. The impact of water stress is neglected in this study ($\gamma_{SM} = 1$). $\gamma_{T\_i}$ and $E_i$ denote the temperature response and the total emission capacity of the vegetation type $i$, respectively. In this study, $\gamma_{T\_i}$ represented the temperature response curves of two types of vegetation, Arctic grass and boreal deciduous shrub, as described in Eqs. (2)-(8). The $LAI$ is normalized by the maximum value of $LAI$ to eliminate the uncertainties of the absolute values of $LAI$, and the ratio between $LAI$ and $LAI_{max}$ depicts the relative change of leaf biomass. The $E_i$ represents the total emission capacity of the vegetation $i$ in the canopy and is as:

$$E_i = \varepsilon_i^* \cdot CF_i \tag{10}$$

$\varepsilon_i^*$ and $CF_i$ represent the canopy-level emission factors and the cover fraction of vegetation $i$, respectively. Canopy-level emission factors were scaled up from the leaf-level emission factors to the scenario with LAI = 5 in MEGANv2.1[13,26]. The leaf-level emission factor for sedges is from the observations for *Eriophorum* spp. in this study at TFS. The leaf-level emission factor for willows is based on the measurements by Wang, et al.[31]. We fitted the model to get the optimal $E_i$ for the default model and $CF_i$ for the updated model using the least-squares method to get rid of the uncertainties associated with vegetation fraction input.

## Isoprene flux measurements

Isoprene flux measurements from three high-latitude sites were used to validate the models. The flux data used in this study were measured in Abisko (Sweden, 68.36° N, 19.05° E) in 2018, Finse (Norway, 60.60° N, 7.53° E) in 2019 and Siikaneva (Finland, 61.83° N, 24.19° E)[20,21,52] in 2021. The campaign times and major vegetation types at the three sites are listed in Supplementary Table 1, and more details about the flux measurements can be found in Seco, et al.[24]. and Vettikkat, et al.[26].

## Community Land Model 5 and numerical experiments

CLM5[43] is the land component of the Community Earth System Model (CESM) and can simulate the land surface and terrestrial ecosystem response and feedback to the weather system and climate change. MEGANv2.1 is coupled to CLM5 as the BVOC emission module. We used CLM5 to test the influence of updating the isoprene temperature response curve on a regional-global scale. We conducted two CLM5 runs: one with the default MEGAN isoprene temperature response and the other with the updated MEGAN using the strong Arctic isoprene temperature responses reported in this study. The emission factors are based on the MEGANv2.1. The model components set is I1850Clm50BgcCrop (https://www.cesm.ucar.edu/models/cesm2/config/compsets.html), and CLM5 is driven by the Global Soil Wetness Project Phase 3 (GSWP3) reanalysis climate forcing dataset with a 3-hour intervals. The spatial resolution of the CLM5 runs is 0.9° × 1.25°, and the numerical experiments cover the period from 1950 to 2009 with the monthly outputs. We adopted the default initial condition of CLM5 that comes from the steady state of the model and treated the first 10 years as the spin-up time. Our analysis is based on the model outputs for the years from 1960 to 2009.

## Data availability

The plant experiment data and model data used in this study are available in the Zenodo database under the accession code: https://zenodo.org/records/11090365.

## Code availability

The code used in this study for analyzing the data and generating figures are available in the Zenodo database under the accession code: https://zenodo.org/records/11090365.

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

## Acknowledgements

H.W., A.W., C.C., and A.G. were supported by the National Science Foundation (NSF) Arctic Natural Sciences (ANS) program award ANS-2041251. R. Sheesley is supported by NSF ANS program award ANS-2041240. K.C.B. was supported by NSF ANS program award ANS-2041250. J.T. is supported by Swedish FORMAS (Forskningsråd för hållbar utveckling) mobility Grant (2016-01580) and Villum Young Investigator (Project no. VIL53048). R. Seco, R.R., J.T., and T.H. were supported by the European Research Council (ERC) under the European Union's Horizon 2020 research and innovation program (grant agreement No. 771012). R.R. and J.T. also acknowledge the support of The Danish National Research Foundation for activities within the Center for Volatile Interactions (VOLT, DNRF168). R. Seco acknowledges a Ramón y Cajal grant (RYC2020-029216-I) funded by MCIN/AEI/ 10.13039/501100011033 and by "ESF Investing in your future", and project PID2021-122892NA-I00 funded by MCIN/AEI and by "ERDF A way of making Europe". IDAEA-CSIC is a Severo Ochoa Center of Research Excellence (MCIN/AEI, Project CEX2018-000794-S). S.S. and L.V. are supported by the Academy of Finland grants No. 337550, No. 346371, and No.310682. The authors are grateful for the support of the staff at the Toolik Field Station, managed by the University of Alaska Fairbanks, and for Dr. Ned Fetcher's advice regarding Arctic plants. We would like to acknowledge high-performance computing support from "Derecho: HPE Cray EX System" (https://doi.org/10.5065/qx9a-pg09) provided by NCAR's Computational and Information Systems Laboratory, sponsored by the National Science Foundation.

## Author contributions

A.G., H.W., and S.N. conceptualized the study. H.W., S.N., A.W., S.B., and A.G. developed the methodology. H.W., A.W., C.L., C.C., and A.G. were responsible for the VOC sample collections. H.W., S.N., and A.G. analyzed the VOC samples. H.W. ran the simulations. H.W. analyzed and visualized the data. J.T., R. Seco, R.R., L.V., S.S., and T.H. provided the flux data. A.G. and C.C. provided supervision throughout the project. H.W., S.N. wrote the first draft of the manuscript. All authors, including R. Sheesley and K.C.B., contributed to the discussion and revision of the manuscript.

## Competing interests

The authors declare no competing interests.
