## [Peer Review File · Nature Communications]

High temperature sensitivity of Arctic isoprene emissions explained by sedgesEditorial note: Parts of this Peer Review File have been redacted as indicated to remove third-party material where no permission to publish could be obtained.

REVIEWER COMMENTS

Reviewer #1 (Remarks to the Author):

The article presents the temperature sensitivity and emission potential of sedges in the Arctic region. The authors highlight that the Arctic sedges have a larger temperature sensitivity, in comparison to other Arctic species e.g. Arctic willow, using environmental gas chamber experiments. The authors used MEGAN to simulate and compare the isoprene fluxes eddy-covariance measurements at 3 high-latitude sites. The results presented are quite interesting and should be useful to the community. The article is well written, and we recommend its publication to Nature Communication, after the authors have addressed the following comments.

1. Why is the 10-day average temperature used for *Carex* spp, since neither the AE nor EF shows a statistically significant value for 5-15 days (Supplementary Figure 4)? I think this is important to address since the emission potential of *Carex* spp is the highest. Therefore, it is important to address the choice of 10-day average temperature for *Carex* spp. What would the impact of, say, a 2-day temperature range be on the *Carex* spp emission potential? Would it still have a higher emission potential compared to *Eriophorum* Spp?

2. Supplementary Figure 4: The right axis is the emission factor, and the left axis is the activation energy.

3. It is difficult to see the differences in the temperature response for Guenther 2012 and Sedges for $T < 30$ C. Please consider replotting it in a way that clearly shows the distinct temperature response curves, including, perhaps, an inset plot. Currently, the differences in the temperature range < 30 oC is not visible.

4. Supplement Table 2: There doesn't seem to be a huge difference in the R2 or RMSE for

the 3 sites, especially Abisko and Finse. A hypothesis or t-test would be useful to assess the significance of the change in RMSE.

5. Figure 2: Why does the default setup predict higher flux in springtime in Finse, even though the only change from the default model is the updated emission factors and temperature response for sedges and Arctic grass? Is it due to updated vegetation fractions? If so, it would be better to state it in the text. The table T1 can also be updated to indicate the old vegetation fractions in the default model case. I suspect the higher isoprene flux in spring is not only due to updated vegetation type; they should also affect the summertime flux emissions. Factoring in Figure 1a, if one considers the temperature response curves for default Megan (Guenther 2012) and the updated emission from the sedges, they appear to be similar for $T < 30$ C, which makes this increase in springtime isoprene flux puzzling.

6. Figure 3: Why is there minimal relative change in isoprene emissions in the northern regions (Norway, Finland and Sweden) where comparisons with the flux measurements were performed? There seems to be really high relative changes in the Russian Siberian region, and it is hard to assess that the changes in the sedges emission potential can have such a huge impact here? How can you justify the changes in the Russian region without adequate comparison to any flux measurement data from the site?

7. How is the inhibition of isoprene by CO₂ treated in the model?

8. Why is the deactivation energy set to 230 KJ/mol? Figure 1b shows that activation energies can exceed 250 KJ/mol at lower 10-day average temperatures.

9. Although the authors claim that isoprene emissions changed by 20% between 2000 and 2009 and speculate on the potential impacts on SOA, O₃, and CH₄, it would be valuable to see the perceived changes in all species concentrations using global model simulations. At least for the SOA formation, which is pointed to increase, a more sensitive study is required. As higher isoprene emission decreases the OH concentration, there will be less SOA formation through the oxidation of monoterpenes. It is well-known that monoterpenes

have a much higher SOA formation potential than isoprene.

Reviewer #1 (Remarks on code availability):

We have not reviewed the code.

Reviewer #2 (Remarks to the Author):

The comments will come from my post-doctoral student, as this was a common review.

Reviewer #2 (Remarks on code availability):

All data were available.

Reviewer #3 (Remarks to the Author):

High-temperature sensitivity of isoprene emissions has been reported in different Arctic ecosystems for the last decade or so. The exact reason is not well known. Such high-temperature sensitivity has many implications for our understanding of plant physiology, high-latitude atmospheric chemistry, and climate systems. This manuscript has contributed to this knowledge gap in two ways. They have identified that sedges, and not all Arctic vegetation, exhibit a very strong temperature response, thus key contributors to some of observed the ecosystem level sensitivity. Using the chamber experiments, this work also explored the mechanism for such high-temperature sensitivity in Arctic plants and used the information to inform an updated MEGAN algorithm to account for it. It shows that the updated MEGAN can simulate the landscape/ecosystem level isoprene fluxes better than the default MEGAN, particularly in those low-emission periods. The results of how differences in MEGAN perditions between the updated and default MEGAN for current and historical emissions are also of high interest.

The update in MEGAN to account for such high-temperature sensitivity in high-latitude vegetation has been expected from the community for a while. I am pleased to see the authors report the update here. I expect it will be incorporated into many other community models and thus will help facilitate more research into this topic. The manuscript is well-

written, and the experiments are well-designed. I don't have any major concerns about the publication of this work. As I said, the updated MEGAN is a much-needed development.

A few minor questions/suggestions:

- Emission factor response to high-temperature sensitivity vs temperature response curve/ γ T: How to attribute some of the sensitivity to the changed emission factor (Arctic willow), while in other species (sedges), it is due to the temperature response curve?
- Has the updated MEGAN changed the emission factors for Arctic vegetation? I was not sure if it had been updated or not.
- There appears to be a bit of repetition between the last paragraph of the 'Abstract' and the first two paragraphs in 'Results and Discussion'.

Reviewer #3 (Remarks on code availability):

There is no README file but the file names seem to be intuitive. I didn't check if the code could be executed.

Reviewer #4 (Remarks to the Author):

This manuscript conducted species-specific chamber experiments, which demonstrate a pronounced temperature response of isoprene emissions from Arctic sedges. This finding sheds light on previous underestimations of the temperature effect on isoprene emissions in high-latitude tundra ecosystems. By updating this temperature response curve for sedges and the dependence of emission capacity on the previous day temperature for willows, the manuscript shows a substantial underestimation in both the magnitude and long-term trend of isoprene emissions in the high-latitude regions. These findings are novel and of significance to improve model simulations of BVOC emissions. I'd recommend accept this manuscript after addressing the following comments.

1. To make the manuscript suitable for Nature Communications, I'd recommend the authors present a more comprehensive introduction of BVOC emissions by emphasizing the interdisciplinary nature of their study. In particular, I'd recommend the authors explain why plants emit isoprene, and biologically why isoprene emissions are sensitive to temperature,

and why there is large uncertainty of the temperature sensitivity of isoprene emissions. I found the discussions about the enzyme activity very interesting, but I don't see any discussions about this in introduction.

2. The manuscript is overall well-structured, but the methodology section could benefit from further clarification to enhance its comprehensibility. For example, in the MEGAN model and temperature response curve section, the authors wrote "The updated short-term temperature response curve for sedges", "long-term temperature response for boreal broadleaf deciduous shrub". The manuscript does not explain the short-term vs. long-term, and the 10-day average temperature is considered as short-term?

3. The temperature sensitivity of isoprene emissions from sedges is mainly derived from short-term experiments, but it's unclear to me whether the short-term temperature sensitivity will hold true at longer time scales. With the global warming, plants may adapt to warmer environment, and their sensitivity to temperature may also change.

4. The manuscript concludes by discussing the likely increased high-latitude isoprene emissions in response to increasing heatwaves and general warming and the impacts on regional chemistry and climate system, but such discussions are most qualitative. To fortify the manuscript's impact, it would be beneficial to delve into the broader implications by quantitatively assessing the impacts of emission changes on global radiative forcing and atmospheric chemistry.

Minor comments:

1. Regarding the leaf chamber experiments: previous studies have suggested a dependence of BVOC emissions on soil moisture, but in the chamber experiments plant samples were detached from soil and submerged into water, how might this affect the results?

2. "Isoprene flux measurements", add one sentence to state what the flux measurements are used for to improve readability.

3. What are the uncertainties on the isoprene emissions by assuming 10-day average temperature in the framework?

4. For Fig. 2 model prediction vs observation, is R^2 similar for cold and warm periods? Why does R^2 vary a lot across different sites?
5. Supplementary Fig. 4: AE on left axis, EF on y axis.

Reviewer #5 (Remarks to the Author):

Response to Reviewers

We have revised the paper according to the reviewers' suggestions and comments. In addition, we added one more corresponding author, Riikka Rinnan. The point-by-point comments are given below.

Reviewer #1:

The article presents the temperature sensitivity and emission potential of sedges in the Arctic region. The authors highlight that the Arctic sedges have a larger temperature sensitivity, in comparison to other Arctic species e.g. Arctic willow, using environmental gas chamber experiments. The authors used MEGAN to simulate and compare the isoprene fluxes eddy-covariance measurements at 3 high-latitude sites. The results presented are quite interesting and should be useful to the community. The article is well written, and we recommend its publication to Nature Communication, after the authors have addressed the following comments.

Response: Thank you so much for your precious time and comments. We are very willing to revise the paper according to your comments and suggestions. The point-by-point comments are given below.

1. Why is the 10-day average temperature used for *Carex* spp, since neither the AE nor EF shows a statistically significant value for 5-15 days (Supplementary Figure 4)? I think this is important to address since the emission potential of *Carex* spp is the highest. Therefore, it is important to address the choice of 10-day average temperature for *Carex* spp. What would the impact of, say, a 2-day temperature range be on the *Carex* spp emission potential? Would it still have a higher emission potential compared to *Eriophorum* Spp?

Response: Thank you so much for your question. Species of the grass-like *Carex* spp. and *Eriophorum* spp. both belong to the Cyperaceae family, and thus we have assumed they would exhibit similar behaviors under changing temperatures until there is more evidence that there is a difference in their behavior. Since we have more measurements from *Eriophorum* spp. (n=9) than from *Carex* spp. (n=5), we had greater confidence in using *Eriophorum* spp. data to construct the model used in this study for grass tundra ecosystems. Additionally, the relatively small number of measurements for *Carex* spp. (n=5) is a contributing factor to why *Carex* spp. did not show statistical significance for the 5-15 day period, even though the correlation coefficient was very close between the 2-day and 5-10 day data. In the main manuscript, the decision to use a 10-day average temperature as the indicator for long-term warming was primarily based on the *Eriophorum* spp. measurements, but this approach may carry some uncertainty. More

rigorous follow-up experiments, with a larger number of replicates in a walk-in environmental chamber, are needed to investigate if there are differences between these genera and to determine the precise timescale for this long-term impact.

In addition, *Carex* spp. did not exhibit a higher emission capacity than *Eriophorum* spp., but *Salix* spp. did (Fig. R1). *Salix* spp., which are isoprene-emitting shrubs in the Arctic, were thoroughly investigated for their temperature response in the Arctic environment in our previous study (Wang et al., 2024). As shown in Fig. R2, the minute-to-hour temperature response of *Salix* spp. closely matches the Model of Emissions of Gases and Aerosols from Nature (MEGAN). However, the emission capacity of *Salix* spp. is highly correlated with the average temperature of the preceding day (Fig. R3). For the modeling component of this study, we utilized the temperature response curve derived from our previous study (Wang et al., 2024) to update the model for boreal shrubs as outlined in the Methods Section.

Fig. R1. Comparison of leaf-level isoprene emissions from vegetation species at the Toolik Field Station. The measurements were conducted when the leaf temperature was about 30°C under a PPFD of 1000 $\mu\text{mol m}^{-2} \text{s}^{-1}$. The green triangle represents the mean, while the orange line represents the median. The upper and lower boundaries of the box represent the first and third quartiles, respectively. The whiskers extend from the box by 1.5 times the inter-quartile range.

Fig. R2. (a) Comparing temperature responses of isoprene emissions between this and previous studies. Short-term temperature response curve of willows (orange solid line) derived through leaf chamber experiments in this study, along with tundra whole-ecosystem measurement response curves from previous studies (various colors and patterns). The orange shadow represents the 95% confidence intervals. GC, BC, and EC denote ground chamber, branch chamber, and eddy-covariance measurements. Curves are normalized to emission at a leaf temperature of 30 °C. (b) Emission factors of different *Salix* spp. in the Arctic. Emission factor is defined as isoprene emission capacity at 30 °C and PPFD of 1000 μmol m⁻² s⁻¹. Averaged emission factors are shown per leaf area (left axis, orange) and per dry leaf mass (right axis, green). Points and error bars represent mean and standard deviation of emission factors. (Wang et al., 2024)

Fig. R3. (a) The time-series of normalized emission factors from different leaves (various colors and patterns, left axis) and daily temperatures (solid orange line, right axis). The blue and red dashed lines represent the mean daily temperature and 95th percentile of the daily temperature records during 2020-2023, respectively. The orange shadow represents the standard deviation of daily temperature in (a). The emission factors were normalized by dividing the mean emission factors for individual leaves. (b) The correlation between normalized emission factors and the previous-day averaged temperature is shown

alongside that in the default MEGAN model (dashed green line). The orange shadow represents the 95% confidence intervals in (b). The equations for the fitted lines in (b) are also presented (Wang et al., 2024)

2. Supplementary Figure 4: The right axis is the emission factor, and the left axis is the activation energy.

Response: Thank you for pointing that out. We have corrected that.

3. It is difficult to see the differences in the temperature response for Guenther 2012 and Sedges for $T < 30$ C. Please consider replotting it in a way that clearly shows the distinct temperature response curves, including, perhaps, an inset plot. Currently, the differences in the temperature range $< 30^\circ\text{C}$ is not visible.

Response: Thank you so much for your suggestion, and it is very helpful! The updated Fig. 1 with the caption at Line 186 is as:

“Fig. 1. (a) presents the temperature responses of isoprene emissions from this and previous studies in the northern high-latitude regions. (b) is the same plot as (a), but only for temperatures under 30°C; (c) shows the relationship between the isoprene temperature sensitivities and emission capacities of sedges. The short-term temperature response curves of sedges up to 35 °C from this study is shown by the orange solid line, and the orange shading represents the 95% confidence intervals. The short-term temperature response curves of tundra ecosystem from previous studies are also presented by lines with different colors and patterns. GC, BC and EC represent ground chamber experiments, branch chamber experiments and eddy-covariance measurements. The temperature response curves are normalized to the emission level when the leaf temperature equals 30 °C. The temperature curves in Tang, et al. ³⁹ and Li, et al. ²² came from the ground chamber

observations of mixed local vegetation at the Abisko site. Li, et al. ²² also did the branch chamber experiments for *Salix myrsinifolia* L. (purple solid line). The site in Seco, et al. ¹⁸(2020) is located in a sedge-dominated fen near the Abisko-Stordalen site. The Abisko measurements in Seco, et al. ²¹ (2022) happened at a different location within the same Abisko-Stordalen area on an ombrotrophic permafrost plateau. The Finse site in Seco, et al. ²¹ (2022) is a tundra with mixture of fen and heath vegetation with shrubs and lichens. The Siikaneva site is in a fen dominated by moss, sedges and dwarf shrubs, and surrounded by Scots pine forest.²³ (c) presents an inverse relationship between the activation energies of the isoprene temperature response and the isoprene emission factors for *Eriophorum* spp. (circle) and *Carex* spp. (triangle). The green dashed line in (b) shows the activation energy in the Model of Emissions of Gases and Aerosols from Nature (MEGAN)¹³. The colors of markers denote the average temperatures over the previous 10 days, and emission factor is defined as the level of isoprene emission at a leaf temperature of 30 °C and a photosynthetic photon flux density (PPFD) of 1,000 $\mu\text{mol m}^{-2} \text{s}^{-1}$.”

4. Supplement Table 2: There doesn't seem to be a huge difference in the R2 or RMSE for the 3 sites, especially Abisko and Finse. A hypothesis or t-test would be useful to assess the significance of the change in RMSE.

Response: Thank you so much for your suggestion. RMSE is a single-number metric, and we cannot apply the t-test to RMSE. Instead, we assess the significance of model error change (mean absolute error, MAE). The results for MAE and the corresponding t-test results have been added to Supplementary Table 2 as follows:

“**Supplementary Tab. 2. The performances of models.** The statistics of the different temperature response curve models at the Abisko-Stordalen, Finse, and Siikaneva sites with the least square fitting. RMSE and MAE are short for the root mean square error and mean absolute error in the unit of $\text{nmol m}^{-2} \text{s}^{-1}$, respectively. T-tests were applied to test the significance between the differences of MAE.

Site	Abisko-Stordalen				Finse				Siikaneva			
	R^2	Slope	RMSE	MAE ($p < 0.05$)	R^2	Slope	RMSE	MAE ($p = 0.22$)	R^2	Slope	RMSE	MAE ($p < 0.01$)
Updated MEGAN v2.1	0.81	0.84	0.45	0.24	0.68	0.74	0.58	0.28	0.90	0.89	1.02	0.87
Default MEGAN v2.1	0.78	0.73	0.48	0.21	0.64	0.62	0.60	0.27	0.83	0.80	1.37	0.64

5. Figure 2: Why does the default setup predict higher flux in springtime in Finse, even though the only change from the default model is the updated emission factors and temperature response for sedges and Arctic grass? Is it due to updated vegetation fractions? If so, it would be better to state it in the text. The table T1 can also be updated to indicate the old vegetation fractions in the default model case. I suspect the higher isoprene flux in spring is not only due to updated vegetation type; they should also affect the summertime flux emissions. Factoring in Figure 1a, if one considers the temperature response curves for default Megan (Guenther 2012) and the updated emission from the sedges, they appear to be similar for $T < 30 \text{ C}$, which makes this increase in springtime isoprene flux puzzling.

Response: Thank you for your question. In the default model simulation, the uncertainties of vegetation cover fraction have been included in the total emission factors (E) as presented in Equation (10) in the Method section of the manuscript. These factors represent the emission capacity of the entire ecosystem (or the footprint of the flux measurements) by considering leaf-level emission factors, biomass, and vegetation cover fraction. Because all environmental factors are consistent across all plant functional types (PFTs), it is not necessary to differentiate the vegetation types from those in the default MEGAN model. Therefore, we can use E to represent the total emission capacity for the area covered by the flux tower footprint.

We fitted the "best" estimation of the emission factor for the Finse site using the least squares fitting method for the observed fluxes over the entire season. After that, we still observed an overestimation of isoprene during the spring, a phenomenon also reported by Seco et al., 2022. We believe this is caused by the inability of the temperature response curves to adequately represent the response to long-term (including seasonal) warming. The emission of isoprene is proportional to the emission factors in the default MEGAN model. As you mentioned in your question, if we aim to capture the high values of isoprene in summer, adjusting the canopy emission factor will lead to an overestimation in colder periods like spring.

In addition, the temperature response curve we proposed is a dynamic temperature response curve relating to the 10-day averaged temperature, and the temperature curve for sedges shown in Fig. 1 was fitted by all the measurements we have. As shown in Fig. R4, the sedges could respond to a cold environment and have a higher activation energy for the temperature response curve, which would lead to a much lower value of γ_T . The 10-day averaged temperature during spring to early summer at the Finse site is from -3.7 to 7.0 °C, so the temperature response curve for the sedges could be significantly lower than that of the default MEGAN model as shown in Fig. R4.

Fig. R4. The temperature curves of sedges under the different 10-day averaged temperature. T240 and AE represents the 10-day averaged temperature and corresponding activation energy, respectively.

As you suggested, we revised the corresponding statement about the emission factors in the revised paper as in the Method section at Line 467:

“We fitted the model to get the optimal E_i for the default model and CF_i for the updated model using the least-squares method to get rid of the uncertainties associated with vegetation fraction input.”

6. Figure 3: Why is there minimal relative change in isoprene emissions in the northern regions (Norway, Finland and Sweden) where comparisons with the flux measurements were performed? There seems to be really high relative changes in the Russian Siberian region, and it is hard to assess that the changes in the sedges emission potential can have such a huge impact here? How can you justify the changes in the Russian region without adequate comparison to any flux measurement data from the site?

Response: Thank you so much for this very good question. This change is likely related to the vegetation distribution and the differences in temperature responses between the Arctic sedges and shrubs as shown in Fig. R5. As mentioned in the main manuscript, we updated the temperature response curves for both Arctic grass and boreal deciduous shrubs plant functional types (PFTs) based on this study and our previous study on *Salix* spp. (Wang et al., 2024). For the sedges (represented by Arctic grass), the new model simulates a decrease in isoprene during the cold period compared to the default model, which was found when we compared the model results with the flux measurements. In that case, the model predicts a general decrease in isoprene emissions for the Arctic grass-dominated area, including Fennoscandia. However, the boreal deciduous shrub-dominated regions, including the Russian Siberian region, show

an increase due to the updated shrub temperature response curves. Direct isoprene emission measurements do not exist for the Russian Siberian region. However, indirect potential evidence is provided by the Ozone Monitoring Instrument (OMI) formaldehyde (HCHO) measurements, and Stavrakou et al. (2018) suggested that the interannual variability of HCHO in Russian Siberia could be explained by the biogenic isoprene emissions. Nevertheless, we believe that more direct measurements are crucial to validate the model and understand the biogenic isoprene emissions in Siberia. We added the discussion we had here in the manuscript at Line 304 as:

“The simulations predict a notable increase in isoprene emissions in the Russian Siberian regions dominated by boreal deciduous shrubs. The model results presented by Stavrakou et al.⁴⁶ suggested that the interannual variability of Ozone Monitoring Instrument (OMI) formaldehyde (HCHO) measurements in Siberia could be explained by biogenic isoprene emissions. However, more in-situ measurements are crucial to validate the model and understand the biogenic isoprene emissions in Siberia.”

Editorial note: figure redacted

Fig. R5. The spatial distribution of cover fraction for the boreal deciduous shrub and Arctic grass in the Community Land Model version 5 are presented in (a) and (b), respectively. The relative change of isoprene emission during 2000-2009 caused by the new temperature response curves is presented in (c).

7. How is the inhibition of isoprene by CO₂ treated in the model?

Response: Thanks you so much for your question. We adopted the constant CO₂ of 284.7 ppmv in the CLM5 for simulating the longterm trend in BVOCs. According to the method proposed by Heald et al. (2009), our estimations for isoprene emission will have a 9.78% overestimation without considering the trend of CO₂ from pre-industrial times to 2009 (Fig. R6). The CO₂ inhibition of isoprene emission is an important but highly uncertain process that was not the focus of this study. For example, Potosnak et al. (2014) demonstrated that the inhibition effect of CO₂ on isoprene could be eliminated at

increased temperature for the tundra dwarf willows and this behavior has been confirmed by recent studies (Sahu et al., 2023). Because of the uncertainties and potential interactions between CO₂ and temperature responses, we treated the CO₂ exactly the same in our experiments, and it will not affect our evaluation of the impact of temperature response curves on isoprene emissions in the high latitude regions.

Fig. R6. The response of isoprene emissions to the CO₂ concentration based on Heald et al. (2009). The impact of CO₂ at 284.7 and 387.6 ppmv levels are labeled by the pink and orange dashed line. The isoprene emission could be 9.78% lower when CO₂ level rises from 284.7 ppmv to 387.6 ppmv.

8. Why is the deactivation energy set to 230 KJ/mol? Figure 1b shows that activation energies can exceed 250 KJ/mol at lower 10-day average temperatures.

Response: The activation and deactivation energies in the Arrhenius-type temperature response function are used to describe the increase and decrease stages of isoprene under different temperatures (Rasulov et al., 2015). The deactivation energy of 230 kJ/mol for the default MEGAN model (Guenther et al., 2006; Guenther et al., 1993) was fitted from chamber experiments for eucalyptus, sweet gum, aspen, and velvet bean. However, in our experiments, we did not observe a decrease in isoprene even though the leaf temperature reached 40 °C, which is beyond the normal leaf temperature observed in high latitudes. In that case, the new algorithm did not contain a deactivation energy term and only had an activation energy term to describe the increase stage of isoprene emission response to temperature.

9. Although the authors claim that isoprene emissions changed by 20% between 2000 and 2009 and speculate on the potential impacts on SOA, O₃, and CH₄, it would be valuable to see the perceived changes in all species concentrations using global model simulations. At least for the SOA formation, which is pointed to increase, a more

sensitive study is required. As higher isoprene emission decreases the OH concentration, there will be less SOA formation through the oxidation of monoterpenes. It is well-known that monoterpenes have a much higher SOA formation potential than isoprene.

Response: We appreciate taking up this topic. As you identified, we didn't quantitatively evaluate the potential atmospheric impacts of the change in isoprene emissions but only briefly discussed this based on literature. We agree that we could expect a decrease in secondary organic aerosols (SOA) caused by the decrease of OH. However, to fully understand the chemistry involved in this change, we need thorough simulation studies to quantitatively evaluate the change. Therefore, in this study, we focus entirely on explaining this high-temperature response curve and its impact on isoprene emissions in high-latitude regions. The evaluation of chemical impact would be our next step as a new independent project. For the revised manuscript, we changed the upper arrow for SOA change in Fig. 5 to a question mark as:

“

Fig. 5. Schematic representation of isoprene emission increase and atmospheric chemical changes induced by the response of sedges and willows to Arctic warming. Warming will increase isoprene

emission from Arctic ecosystems. This change in isoprene emissions due to warming can alter tropospheric chemistry, resulting in extended methane lifetime, altered tropospheric ozone concentration, and changed aerosol formation. These changes can influence the local radiation energy balance and exacerbate climate fluctuations.”

Reviewer #1 (Remarks on code availability):

We have not reviewed the code.

Response: Data and code are open access through the Zenodo platform: <https://zenodo.org/records/11090365>. We updated the code package with a README file.

Reviewer #2:

The comments will come from my post-doctoral student, as this was a common review.

Response: Thank you so much for your precious time and efforts.

Reviewer #2 (Remarks on code availability):

All data were available.

Response: Data and code are open access through the Zenodo platform: <https://zenodo.org/records/11090365>. We updated the code package with a README file.

Reviewer #3:

High-temperature sensitivity of isoprene emissions has been reported in different Arctic ecosystems for the last decade or so. The exact reason is not well known. Such high-temperature sensitivity has many implications for our understanding of plant physiology, high-latitude atmospheric chemistry, and climate systems. This manuscript has contributed to this knowledge gap in two ways. They have identified that sedges, and not all Arctic vegetation, exhibit a very strong temperature response, thus key contributors to some of observed the ecosystem level sensitivity. Using the chamber experiments, this work also explored the mechanism for such high-temperature sensitivity in Arctic plants and used the information to inform an updated MEGAN algorithm to account for it. It shows that the updated MEGAN can simulate the landscape/ecosystem level isoprene fluxes better than the default MEGAN, particularly in those low-emission periods. The results of how differences in MEGAN perditions between the updated and default MEGAN for current and historical emissions are also of high interest.

The update in MEGAN to account for such high-temperature sensitivity in high-latitude vegetation has been expected from the community for a while. I am pleased to see the

authors report the update here. I expect it will be incorporated into many other community models and thus will help facilitate more research into this topic. The manuscript is well-written, and the experiments are well-designed. I don't have any major concerns about the publication of this work. As I said, the updated MEGAN is a much-needed development.

Response: Thank you for taking the time to review our manuscript and for your insightful comments. We are pleased to hear that you find our work on the high-temperature sensitivity of isoprene emissions in Arctic ecosystems and the updates to the MEGAN algorithm to be a valuable contribution to the field. The point-by-point responses are given below.

A few minor questions/suggestions:

- Emission factor response to high-temperature sensitivity vs temperature response curve/ γ T: How to attribute some of the sensitivity to the changed emission factor (Arctic willow), while in other species (sedges), it is due to the temperature response curve?

Response: Thank you for your question. We conducted individual experiments to distinguish the temperature response of willows and sedges. In our previous study (Wang et al., 2024) on Arctic willows (*Salix* spp.), we found that the minute-to-hour temperature response curve is very close to the MEGAN model. In contrast, the isoprene emission factors of willows exhibited a more substantial than expected response to the mean ambient temperature of the previous day, as shown in Fig. R3 above. In that study, we tried to explain the high temperature sensitivity with the change of emission factor of willows; however, the change in willow emission factors cannot fully explain the high temperature response curves observed in other whole ecosystem measurements, as shown in Fig. R7.

The emission factors and the temperature response curves of sedges both respond to the 10-day average temperature as shown in Fig. R8. During a cold period, the sedges exhibit a strong temperature response curve, and emissions could increase significantly. While going through a warm period, sedges would be less sensitive to minute-to-hour temperature changes as indicated by a lower activation energy, while the emission factors would increase correspondingly. Integrating these two behaviors, the whole ecosystem is likely to exhibit very low isoprene emissions during the cold period, and emissions could increase rapidly with the rise in temperature, reaching a high level after a warming period.

The evidence of different patterns can also be found in ecosystem-level measurements. As shown in Fig. R9 (Seco et al., 2022), the temperature curves derived from the Abisko and Finse sites differ from each other, even though they are both high-latitude tundra

sites. The potential explanation is that the isoprene emission at Finse is dominated by sedges and willows, while at Abisko, it is dominated only by sedges. In that case, the two sites exhibit two different high-temperature response curves.

Fig. R7. The time-series of simulated isoprene emissions from the default (dashed green line, left axis) and updated MEGAN models (solid orange line, left axis), alongside air temperature (dashed pink line, right axis), during the heatwave period (July 16–August 1) in 2023 at the Toolik Field Station (a). The diurnal cycle of simulated isoprene emissions shown in (a) is depicted in (b), and the shadows in (b) represents standard deviations of isoprene emission. (c) Presented the short-term temperature response curve of isoprene flux, obtained from flux estimations using both the default and updated MEGAN models. The equations for the fitted lines in (c) are also presented (Wang et al., 2024).

Fig. R8. The response curves of the temperature sensitivity and emission factor to the past 10-day average air temperature. (a) and (b) present the relationship between the activation energy or temperature sensitivity to the past 10-day average temperature for *Eriophorum* spp. (blue) and *Carex* spp. (orange). (c) and (d) depict emission factors versus the past 10-day average temperature for *Eriophorum* spp. (blue) and *Carex* spp. (orange). The fitted equation and R^2 are both presented.

Editorial note: figure redacted

Fig. R9. (A and C) Measured (square symbols) and (B and D) modeled (circle symbols) isoprene temperature activity factors (γ_T) plotted against the measured vegetation surface temperature for Finse, Norway (A and B), and Abisko, Sweden (C and D) (Seco et al., 2022).

- Has the updated MEGAN changed the emission factors for Arctic vegetation? I was not sure if it had been updated or not.

Response: Thank you for your question. We updated the leaf-level emission factors for our site-scale simulations and fit the vegetation fraction to the measured landscape-average emission factors. The leaf-level emission factors were $12.0 \text{ nmol m}^{-2} \text{ s}^{-1}$ for sedges and $6.5 \text{ nmol m}^{-2} \text{ s}^{-1}$ for the willows, respectively, which were based on our glass chamber measurements for this study and Wang et al. (2024). As shown in Tab. 1 in the manuscript, the estimated vegetation cover fractions were within the expected range for these sites indicating that our estimations of emission factors for sedges and willows are in a reasonable range.

For the CLM5 experiments, we used the default PFT-level emission factors from MEGAN to assess the impact of temperature response curves. Seco et al. (2022) report that the emission factor for the Arctic grass PFT in MEGANv2.1 closely matches the landscape-scale flux measurements, suggesting that about 10% of the vegetation,

specifically sedges, on the tussock tundra are isoprene emitters. However, for boreal deciduous shrubs, emission factors vary depending on the species, as not all deciduous shrubs emit isoprene, according to our measurements. In this case, a detailed species composition distribution map would be required to determine the accuracy of the emission factors for boreal deciduous shrubs. Substantial efforts are required to precisely map the emission factors for deciduous shrubs in high-latitude ecosystems.

- There appears to be a bit of repetition between the last paragraph of the 'Abstract' and the first two paragraphs in 'Results and Discussion'.

Response: Thank you for pointing this out. We removed the repetitive part in the first paragraph in "Results and Discussion", and the current first two paragraphs in "Results and Discussion" at Line 88 are now:

"Our species-level gas exchange chamber experiments showed that the main isoprene emitters among the species measured at Toolik Field Station (TFS), Alaska, USA, are sedges, a major component of Arctic graminoid plants, and willows, a major component of Arctic woody plants (Supplementary Fig. 1). Other studies have also indicated that the *Salix* spp., *Carex* spp. and *Eriophorum* spp. exhibit significantly higher isoprene emission levels compared to other tundra species, e.g., *Betula* spp. and *Cassiope* spp.^{25,29,31-33} and *Sphagnum* spp.³⁴⁻³⁶. The temperature response of willows in the Arctic cannot explain the high temperature sensitivity of isoprene emissions from high-latitude ecosystems³⁰.

Arctic sedges studied here show a more pronounced temperature response than other plant species, including Arctic woody willow shrubs and any of the plant responses used to develop the MEGAN model. Our data confirm that the sedges are responsible for the heightened temperature responses of isoprene emissions from high-latitude ecosystems (Fig. 1a). We calculated the Q10 coefficient for isoprene emissions from sedges (*Carex* spp. and *Eriophorum* spp.) and willows between 25 and 35 °C. The Q10 coefficient represents the isoprene emission rate change with a 10 °C rise of the leaf temperature. The Q10 values of *Carex* spp. (15.6 ± 8.8) and *Eriophorum* spp. (9.1 ± 7.0) are much higher than the Q10 of the Arctic willows (3.2 ± 1.8), which is close to the Q10 of the MEGAN model (2.91) (Supplementary Fig. 2). We applied the Arrhenius equation to model the exponential temperature response curves of *Carex* spp. and *Eriophorum* spp. (refer to the Methods section), where the activation energy (Eq. 5) denotes the temperature sensitivity of isoprene emission. Our findings indicate that the temperature response curves of both *Carex* spp. and *Eriophorum* spp. exhibit high temperature sensitivity (or high activation energy) up to 35 °C (Supplementary Fig. 3). However, the activation energy and R² decrease beyond 40 °C, suggesting a slower increase rate of isoprene emissions from both species (Supplementary Fig. 3)."

Reviewer #3 (Remarks on code availability):

There is no README file but the file names seem to be intuitive. I didn't check if the code could be executed.

Response: Data and code are open access through the Zenodo platform: <https://zenodo.org/records/8339419>. We updated the code package with a README file.

Reviewer #4:

This manuscript conducted species-specific chamber experiments, which demonstrate a pronounced temperature response of isoprene emissions from Arctic sedges. This finding sheds light on previous underestimations of the temperature effect on isoprene emissions in high-latitude tundra ecosystems. By updating this temperature response curve for sedges and the dependence of emission capacity on the previous day temperature for willows, the manuscript shows a substantial underestimation in both the magnitude and long-term trend of isoprene emissions in the high-latitude regions. These findings are novel and of significance to improve model simulations of BVOC emissions. I'd recommend accept this manuscript after addressing the following comments.

Response: Thank you for your encouraging comments on our manuscript and for recognizing the value of our research. We have addressed the points you've mentioned point-by-point.

1. To make the manuscript suitable for Nature Communications, I'd recommend the authors present a more comprehensive introduction of BVOC emissions by emphasizing the interdisciplinary nature of their study. In particular, I'd recommend the authors explain why plants emit isoprene, and biologically why isoprene emissions are sensitive to temperature, and why there is large uncertainty of the temperature sensitivity of isoprene emissions. I found the discussions about the enzyme activity very interesting, but I don't see any discussions about this in introduction.

Response: Thank you for your suggestion. We added one paragraph to better introduce plant isoprene emissions in the Introduction section at Line 46:

"Isoprene is the most abundant reactive BVOC emitted globally and in the Arctic^{3,13,14}. Isoprene can help vegetation to tolerate abiotic stresses¹⁵, and isoprene can act as a signaling compound to stimulate plant defense mechanisms during stress periods¹⁶. Isoprene is synthesized from dimethylallyl diphosphate (DMADP) derived from the methyl erythritol 4-phosphate (MEP) pathway through the enzyme isoprene synthase (IspS)¹⁷. Isoprene emission is controlled by environmental conditions, especially temperature and solar radiation¹⁸. Thus, a rapidly warming climate in the Arctic is favorable for increasing the emission of isoprene¹⁹⁻²². The temperature response curves of isoprene emission, used in the current earth system models (ESMs) and the chemistry transport models (CTMs), are based on measurements of a few temperate

plants^{13,23}, and a typical isoprene temperature response curve has a Q10 of about 3 which is thought to be driven by the influence of temperature on substrate supply and the activity of IspS¹⁷. However, recent whole-ecosystem measurements suggest that the temperature response of isoprene emissions in high-latitude tundra ecosystems has a Q10 over 8, which is also much higher than that predicted by the widely used BVOC emission model, the Model of Emissions of Gases and Aerosols from Nature (MEGAN)^{21,24-28}.”

2. The manuscript is overall well-structured, but the methodology section could benefit from further clarification to enhance its comprehensibility. For example, in the MEGAN model and temperature response curve section, the authors wrote “The updated short-term temperature response curve for sedges”, “long- term temperature response for boreal broadleaf deciduous shrub”. The manuscript does not explain the short-term vs. long-term, and the 10-day average temperature is considered as short-term?

Response: Thank you for your suggestion. The short-term refers to the minute-to-hour temperature response in this manuscript, and long-term impact refers to the influence of 1-10 day average temperatures. We agree that more explanation is needed, so we revised the methods section at Line 425 as:

“MEGAN considers the BVOC responses to the long-term temperature, defined as the temperature of the past one day or longer, and the current temperature, reflecting changes on a minute-to-hour scale. The default short-term temperature response curve, γ_T , for isoprene in MEGAN is...”

3. The temperature sensitivity of isoprene emissions from sedges is mainly derived from short-term experiments, but it’s unclear to me whether the short-term temperature sensitivity will hold true at longer time scales. With the global warming, plants may adapt to warmer environment, and their sensitivity to temperature may also change.

Response: Thank you for your comment. Our results address the question of short-term temperature responses of sedges. When we conducted the short-term temperature curve experiments, our sedge samples went through periods with varying temperatures. As shown in Fig. 1(b), the activation energy of the temperature response curve for sedges would decrease with a higher 10-day average temperature. This phenomenon suggests that the isoprene emissions from sedges will be less sensitive to short-term temperature changes and would be similar to those of temperate plants after a warming period, which is represented in the MEGAN model. This aligns with your suggestion that the sensitivity of sedge isoprene emissions to temperature could change in response to the general warming and cooling of the climate on approximately a 10-day timescale. However, we also note that the emission capacity of these sedges could increase when their sensitivity decreases. In that case, we would still expect increased isoprene emissions in a warming climate.

4. The manuscript concludes by discussing the likely increased high-latitude isoprene emissions in response to increasing heatwaves and general warming and the impacts on regional chemistry and climate system, but such discussions are most qualitative. To fortify the manuscript's impact, it would be beneficial to delve into the broader implications by quantitatively assessing the impacts of emission changes on global radiative forcing and atmospheric chemistry.

Response: Thank you for your suggestion. We agree that a quantitative assessment of the atmospheric chemistry impacts of isoprene changes would be valuable next step. However, it is beyond the scope of this study to design and conduct the quantitative simulations needed to investigate the impact of warming on atmospheric chemistry in high-latitude regions. For this study, we focus on reporting our chamber experiments and isoprene emission modeling.

Minor comments:

1. Regarding the leaf chamber experiments: previous studies have suggested a dependence of BVOC emissions on soil moisture, but in the chamber experiments plant samples were detached from soil and submerged into water, how might this affect the results?

Response: Thank you for your question. The water stress that is reflected in soil moisture could affect isoprene emission. If the plant is in a stressed situation, isoprene emission could be affected by changes in leaf temperature and substrate supply, which have been investigated in our previous studies (Seco et al., 2015; Wang et al., 2022). For this study, we didn't observe the behavior of drought when we collected the vegetation samples. We also didn't see any inhibition, or other changes, of photosynthesis behavior when conducting our experiments, which indicates that the plants were not stressed.

2. "Isoprene flux measurements", add one sentence to state what the flux measurements are used for to improve readability.

Response: Thank you for your suggestion. We added one sentence as:

"Isoprene flux measurements from three high-latitude sites were used to validate the models."

3. What are the uncertainties on the isoprene emissions by assuming 10-day average temperature in the framework?

Response: Thank you for your question. We can qualitatively see that the behavior of sedges is related to long-term temperature changes as shown in Fig. R10. However, we do not think our experiments can accurately determine the exact timescale for this behavior, and our adoption of the 10-day average temperature is based on the current

model framework and our experience with *Eriophorum* spp. We expect to conduct more experiments in the future to quantitatively investigate this acclimation behavior.

Fig. R10. Correlation coefficients of the activation energy (left axis) and the emission factors (right axis) of sedges with the mean temperature during the 1 to 15 days preceding the measurement. (a), (b), and (c) display the Pearson correlation coefficients for the activation energy (AE, in blue) and emission factors (EF, in orange) in relation to the mean temperature of the preceding 1 to 15 days for *Eriophorum* spp., *Carex* spp., and a combined analysis of both species, respectively. Statistically significant correlation coefficients ($p < 0.05$) are indicated by solid filled points.

4. For Fig. 2 model prediction vs observation, is R^2 similar for cold and warm periods? Why does R^2 vary a lot across different sites?

Response: We defined the cold and warm periods based on a leaf temperature threshold of 20°C. As shown in Fig. R11, the updated model shows better performance than the default MEGAN model. We observed a difference in model performance between different sites. The model also performs better at the Abisko and Siikaneva sites in general, and the updated model shows much greater improvement at the Finse and Siikaneva sites than at the Abisko site. We speculate that the difference in R^2 is related to the heterogeneous landscapes at these sites. Not all ecosystems have homogeneous distributions of isoprene emitters (Seco et al., 2020), but our model assumes a homogeneous source of isoprene emission. The primary isoprene source is sedges at the Abisko and Siikaneva sites, but at the Finse site, it includes both sedges and willows. In that case, a heterogeneous source of isoprene at the Finse site will affect the flux measurements and the model simulations."

Abisko-Stordalen 2018

Finse 2019

Siikaneva 2021

Fig. R11. The comparison of the measured and simulated isoprene fluxes from the default (blue) and updated MEGAN models (pink) for the Abisko, Finse, and Siikaneva sites during the cold (a, c, e) and warm (b, d, f) periods is shown. The cold and warm periods are defined by a threshold temperature of 20 °C.

5. Supplementary Fig. 4: AE on left axis, EF on y axis.

Response: Thank you for your comment. We have made the correction.

Reviewer #5:

Response: Thank you for accepting the review invitation for our manuscript. We have addressed the comments from the co-reviewer.

Reference

- Guenther, A., Karl, T., Harley, P., Wiedinmyer, C., Palmer, P., and Geron, C.: Estimates of global terrestrial isoprene emissions using MEGAN (Model of Emissions of Gases and Aerosols from Nature), *Atmos. Chem. Phys*, 6, 3181-3210, 2006.
- Guenther, A. B., Zimmerman, P. R., Harley, P. C., Monson, R. K., and Fall, R.: Isoprene and monoterpene emission rate variability: Model evaluations and sensitivity analyses, *Journal of Geophysical Research: Atmospheres*, 98, 12609-12617, doi:10.1029/93JD00527, 1993.
- Heald, C. L., Wilkinson, M. J., Monson, R. K., Alo, C. A., Wang, G., and Guenther, A.: Response of isoprene emission to ambient CO₂ changes and implications for global budgets, *Global Change Biology*, 15, 1127-1140, <https://doi.org/10.1111/j.1365-2486.2008.01802.x>, 2009.
- Potosnak, M. J., LeSturgeon, L., and Nunez, O.: Increasing leaf temperature reduces the suppression of isoprene emission by elevated CO₂ concentration, *Science of The Total Environment*, 481, 352-359, <https://doi.org/10.1016/j.scitotenv.2014.02.065>, 2014.
- Rasulov, B., Bichele, I., HÜVe, K., Vislav, V., and Niinemets, Ü.: Acclimation of isoprene emission and photosynthesis to growth temperature in hybrid aspen: resolving structural and physiological controls, *Plant, Cell & Environment*, 38, 751-766, <https://doi.org/10.1111/pce.12435>, 2015.
- Sahu, A., Mostofa, M. G., Weraduwege, S. M., and Sharkey, T. D.: Hydroxymethylbutenyl diphosphate accumulation reveals MEP pathway regulation for high CO₂-induced suppression of isoprene emission, *Proceedings of the National Academy of Sciences*, 120, e2309536120, 10.1073/pnas.2309536120, 2023.
- Seco, R., Karl, T., Guenther, A., Hosman, K. P., Pallardy, S. G., Gu, L., Geron, C., Harley, P., and Kim, S.: Ecosystem-scale volatile organic compound fluxes during an extreme drought in a broadleaf temperate forest of the Missouri Ozarks (central USA), *Global Change Biology*, 21, 3657-3674, doi:10.1111/gcb.12980, 2015.

Seco, R., Holst, T., Matzen, M. S., Westergaard-Nielsen, A., Li, T., Simin, T., Jansen, J., Crill, P., Friborg, T., Rinne, J., and Rinnan, R.: Volatile organic compound fluxes in a subarctic peatland and lake, *Atmos. Chem. Phys.*, 20, 13399-13416, 10.5194/acp-20-13399-2020, 2020.

Seco, R., Holst, T., Davie-Martin, C. L., Simin, T., Guenther, A., Pirk, N., Rinne, J., and Rinnan, R.: Strong isoprene emission response to temperature in tundra vegetation, *Proceedings of the National Academy of Sciences*, 119, e2118014119, 10.1073/pnas.2118014119, 2022.

Stavrakou, T., Müller, J. F., Bauwens, M., Smedt, I., Roozendaal, M., and Guenther, A.: Impact of Short-term Climate Variability on Volatile Organic Compounds Emissions Assessed Using OMI Satellite Formaldehyde Observations, *Geophysical Research Letters*, 0, 10.1029/2018GL078676, 2018.

Wang, H., Lu, X., Seco, R., Stavrakou, T., Karl, T., Jiang, X., Gu, L., and Guenther, A. B.: Modeling Isoprene Emission Response to Drought and Heatwaves Within MEGAN Using Evapotranspiration Data and by Coupling With the Community Land Model, *Journal of Advances in Modeling Earth Systems*, 14, e2022MS003174, <https://doi.org/10.1029/2022MS003174>, 2022.

Wang, H., Welch, A., Nagalingam, S., Leong, C., Kittitanuvong, P., Barsanti, K. C., Sheesley, R. J., Czimczik, C. I., and Guenther, A. B.: Arctic Heatwaves Could Significantly Influence the Isoprene Emissions From Shrubs, *Geophysical Research Letters*, 51, e2023GL107599, <https://doi.org/10.1029/2023GL107599>, 2024.

REVIEWERS' COMMENTS

Reviewer #1 (Remarks to the Author):

The response to the review provided by the authors was satisfactory. I recommend the publication of this work in Nature communications.

Reviewer #2 (Remarks to the Author):

Based on our comments, the authors have improved the manuscript, and we recommend publishing it in its current state.

Reviewer #4 (Remarks to the Author):

The authors have addressed most of the reviewers' comments. The only comment I have is about the time scales of the temperature response. Fig. 5 talks about long-term warming, but the reply to Q3 of Reviewer 4 indicates this study addresses short-term changes of temperature response. I'd suggest the authors better clarify this in the manuscript.

Reviewer #5 (Remarks to the Author):

I co-reviewed this manuscript with one of the reviewers.

Response to Reviewers

Reviewer #1:

The response to the review provided by the authors was satisfactory. I recommend the publication of this work in Nature Communications.

Response: Thank you for your positive feedback and recommendation. We appreciate your support.

Reviewer #2:

Based on our comments, the authors have improved the manuscript, and we recommend publishing it in its current state.

Response: Thank you for your feedback and recommendation. We appreciate your support and are glad our revisions meet your approval.

Reviewer #4:

The authors have addressed most of the reviewers' comments. The only comment I have is about the time scales of the temperature response. Fig. 5 talks about long-term warming, but the reply to Q3 of Reviewer 4 indicates this study addresses short-term changes of temperature response. I'd suggest the authors better clarify this in the manuscript.

Response: Thank you so much for your comments. We think it is a good point, and the long-term warming here refers to a 1-2 week scale change in temperature for us, which is addressed in this study. We agree that we didn't address the change in temperature on a scale of months to years in this study, which can also be called long-term warming. In that case, we revised our discussion in the manuscript on page 4 as follows:

“The isoprene emitters, including sedges and willows³⁰, would respond to both short-term, intense heatwaves and long-term warming from minutes to weeks scales by increasing their isoprene emissions (Fig. 5).”

Reviewer #5:

I co-reviewed this manuscript with one of the reviewers.

Response: Thank you for your collaborative review. We appreciate the thorough evaluation and feedback from both reviewers.